# Predicting the Output of Solar Photovoltaic Panels in the Absence of Weather Data Using Only the Power Output of the Neighbouring Sites

**DOI:** 10.3390/s23073399

**Published:** 2023-03-23

**Authors:** Heon Jeong

**Affiliations:** Department of Fire Service Administration, Chodang University, Muan-gun 58530, Republic of Korea; hjeong@cdu.ac.kr; Tel.:+82-61-450-1229

**Keywords:** solar photo voltaic, times series forecasting, meteorology, recurrent neural network, gated recurrent unit, long-short term memory, transformers

## Abstract

There is an increasing need for capable models in the forecast of the output of solar photovoltaic panels. These models are vital for optimizing the performance and maintenance of PV systems. There is also a shortage of studies on forecasts of the output power of solar photovoltaics sites in the absence of meteorological data. Unlike common methods, this study explores numerous machine learning algorithms for forecasting the output of solar photovoltaic panels in the absence of weather data such as temperature, humidity and wind speed, which are often used when forecasting the output of solar PV panels. The considered models include Long Short-Term Memory (LSTM), Gated Recurrent Unit (GRU), Recurrent Neural Network (RNN) and Transformer. These models were used with the data collected from 50 different solar photo voltaic sites in South Korea, which consist of readings of the output of each of the sites collected at regular intervals. This study focuses on obtaining multistep forecasts for the multi-in multi-out, multi-in uni-out and uni-in uni-out settings. Detailed experimentation was carried out in each of these settings. Finally, for each of these settings and different lookback and forecast lengths, the best models were also identified.

## 1. Introduction

The popularity of solar photovoltaic (PV) panels as a source of renewable energy has been at an all time high [1]. The sky-high demand for electricity and the environmental effects of non-renewable energy sources have made energy sources such as solar more desirable [2]. This surge in the number of PV systems installed globally escalates the need for optimization of the performance and minimization of the cost of these systems. For optimization and cost minimization of PVs, forecasting the output of these systems is crucial.

Weather conditions, specially solar radiance, temperature, humidity, and wind speed, highly affect the output of the PV sites [3]. These weather factors are also important in forecasting the output of PV sites. So, it is comprehensible that most of the studies related to forecasting the output of the PV sites focus on weather data [4,5]. However, there may be instances in which weather data are not available. An issue in data collection due to faulty sensors or equipment, power outages and network failures can also result in missing or inaccurate data. Meteorological stations may also be located at a significantly larger distance from solar PV sites, which might cause inaccurate representation of the local weather conditions. Additionally, historical meteorological data may not be available, making it laborious to perform long-term forecasting. Furthermore, the PV sites may be located at remote sites where meteorological stations may be difficult to operate due to the limited infrastructure [6]. Finally, collection and the processing of the meteorological data may be expensive and infeasible for small-scale PV sites with low budgets for operation [7].

### 1.1. Time Series Forecasting

A time series analysis has been a long-standing challenge and has been widely investigated in the past. A range of traditional methods have been developed to tackle the problem, including Hidden Markov Models [8]; Kalman filters [9]; and statistical methods such as ARIMA [10], exponentially weighted moving averages [11,12] and vector auto-regressors [13]. Deep learning techniques have also been applied to time series forecasting (TSF) with Recurrent Neural Networks (RNNs) being the predominant architecture due to their ability to model sequential data [14,15,16,17]. Recently, temporal convolutional networks (TCNs) have gained popularity in the field. In combination with RNNs, graph neural networks (GNNs) have also been used to capture spatial and temporal patterns in data [18,19,20,21,22,23]. With the advent of transformer [24] architectures, RNNs have been replaced in many applications of sequential modeling. The self-attention mechanism has been a key factor in the success of transformers, although the quadratic computation and memory complexity associated with self-attention is problematic for longer sequences. Consequently, the focus of most transformer architectures has been on developing more efficient designs that employ sparser query matrices to compute self-attention [25,26]. Recent developments in deep learning for time series forecasting have integrated classical concepts with modern deep learning techniques, achieving promising results [27,28]. For instance, Autoformer decomposes the original time series into seasonality and trend components and then extracts the dependencies using an autocorrelation block [27]. SCINet employs a multi-resolution analysis approach that combines downsampling techniques, convolutions and a unique interaction block to capture the dependencies in the data [28].

### 1.2. Forecasting of Solar PV Power

There has been immense interest on forecasting the output of solar PV plants. The methods used range from traditional statistical models to simple machine learning models and the latest deep-learning-based techniques.

Most of the studies dealing with the forecast of the output of the solar PV plants utilize the meteorological data related to the respective sites. A study compared various regression techniques ranging from linear least squares to support vector machines (SVM) with different kernel functions in weather forecasts to predict hourly solar intensity [29]. Similarly, another work studied SVM models while differing in their dimensionality reduction techniques for forecasting PV power [30]. A simple neural network has also previously been suggested to forecast the global horizontal irradiance (GHI) and direct normal irradiance (DNI) using weather forecasts as predictors [31]. They utilized the Genetic Algorithm (GA) and Gamma test for initial feature selection and utilized various neural network structures to perform forecasting. A different study used only the endogenous variables to obtain forecasting models such as Autoregressive Integrated Moving Average (ARIMA), k-Nearest-Neighbours (KNN), Neural Networks and neural networks optimized by Genetic Algorithms to predict hourly PV generation with a prediction horizon up to 2 h [32]. A comparison between various statistical and machine learning models such as SVM, Binary regression Trees, Random Forest (RF), gradient boosted regression trees (GRBT) and Generative Additive Models (GAM) was presented for performing 1 day ahead hourly PV power generation forecasts for some powerplants in France [33,34]. The influence of spatial and temporal information on solar power generation was studied in a different work using gradient boosting alongside vector autoregressive model and compared it with the autoregressive model [35]. Probabilistic forecasting of solar power generation has also been performed using quantile regression forests [34,36].

Taking these challenges into account, it is essential that substitute propositions are studied for the forecasting of the output power of PV sites in the absence of meteorological data. In this study, the feasibility of utilizing the output power readings of multiple solar PV sites located near one another, for multi step forecasting of the output power of these sites, was studied. The forecasts were performed in various settings: (i) utilizing the data of a single site for both input and output (univariate input and univariate output), (ii) utilizing the data of multiple sites as the input and that of a single site as the output (multivariate input and univariate output), and (iii) utilizing the data of multiple sites for both input and output (multivariate input and multivariate output). Numerous machine learning models were used to perform these forecasts, including Long Short-Term Memory (LSTM) [37], Recurrent Neural Network (RNN) [38], Gated Recurrent Unit (GRU) [39] and Transformer [24]. The data used for this study came from 41 solar PV sites located in Suncheon, South Korea. It included the readings of the output of the panels recorded at a regular interval of 15 min, over a period of 6 months,

The main contributions of this paper include the following:A study of the feasibility of forecasting solar PV outputs in the absence of meteorological data.Utilizing popular deep learning models for the forecast of the solar PV output for optimization of the performance and minimization of the maintenance costs of PV sites.Identifying an appropriate method for the forecast of solar PV output at various forecasting lengths.Suggesting a suitable workflow for the forecasting of solar PV outputs in scenarios where meteorological data are unavailable and under three different settings: the multivariate, univariate and multi-in uni-out settings.

## 2. Methods

The initial steps of this study were data collection and pre-processing. The data obtained from these steps are defined in Section 3.1. After obtaining the data, the next step prior to making forecasts was training the forecasting models. The models used for training are described in Section 3.2. Finally, the trained models were tested on the unseen data, and the models performing the best were deployed. The results obtained from this step are described in Section 4. All of these steps can be visualized in Figure 1. A view of the location of some PV sites used in this study is shown in Figure 2.

### 2.1. Data Description

The data used in this study came from 50 solar photovoltaic (PV) sites located in Suncheon, South Korea. The data included readings of the output of the PV panels taken at regular intervals of 15 min over a period of 6 months from 1 January 2020 to 31 July 2020. The data were collected in a time series format, with readings taken every 15 min. A total of 11,142 data points were used. A small portion of the used dataset has been shown in the Table 1.

The obtained data were pre-processed, and the first pre-processing step was dealing with missing data points and removing inconsistencies. For example, during nighttime, there should not be any PV output at all, so the data during night time were set to zero if otherwise found. The missing data were also replaced by the average of nearby data points. Similarly, it was possible that poor results could be obtained if the dataset was inconsistent, so, if for any sites it was found that a huge chunk of data was missing, then such sites were discarded. Originally, the dataset contained readings from 50 different sites. However, during the preprocessing step, the data from 10 of the sites were removed due to the presence of inconsistencies and huge intervals of missing data. After this, the data were split into train, test, and validation data and scaled.

It is important to note that this study does not utilize weather data, such as temperature, humidity and wind speed, which are often used in forecasting the output of solar PV panels. This highlights the need for alternative approaches when weather data are not available.

In this study, I focus on multi-step time series forecasting, where the goal is to predict the output of the PV panels based on the readings at multiple time steps in the past. Despite the absence of weather data, this study aims to demonstrate the feasibility of using alternative approaches to forecast the output of solar PV panels.

### 2.2. Used Forecasting Models

For the purpose of this study, I used four popular machine learning models to perform forecasting of the solar PV output. The used models are Recurrent Neural Network (RNN) [38], Gated Recurrent Unit (GRU) [39], Long Short-Term Memory (LSTM) [37], and Transformer [24].

#### 2.2.1. Recurrent Neural Network (RNN) [38]

A Recurrent Neural Network (RNN) is a type of artificial neural network that is well suited for processing sequential data. RNNs are used for tasks such as natural language processing and speech recognition, where the input data have a temporal structure. RNNs maintain an internal hidden state that can capture information from the entire sequence of inputs up to a given time step. This allows RNNs to model long-term dependencies in sequential data.

RNNs consist of a series of interconnected nodes or neurons that are connected in a feedforward manner. The hidden state of the network is updated at each time step based on the previous hidden state and the input data at that time step. This allows RNNs to model the dependencies between the data at different time steps, which is crucial for processing sequential data.

The main drawback of traditional RNNs is that they are prone to the vanishing gradient problem, where the gradient of the error signal with respect to the network parameters decreases exponentially as it propagates through time. This makes it difficult to train RNNs on long sequences, as the gradient of the error signal becomes very small, making it difficult to update the network parameters. To overcome this, various variants of RNNs have been developed, such as LSTMs (Long Short-Term Memory) and GRUs (Gated Recurrent Units), which are more robust to the vanishing gradient problem. The architecture of a general RNN is shown in Figure 3.

The equation for an RNN model at time step t can be expressed as follows:(1)$$h_{t}=f\left(W_{hx}x_{t}+W_{hh}h_{t-1}+b_{h}\right)$$
(2)$$y_{t}=g\left(W_{yh}h_{t}+b_{y}\right)\;$$
where $h_{t}$ is the hidden state at time step *t*, $x_{t}$ is the input at time step *t*, $y_{t}$ is the output at time step *t*, $W_{hx}$ and $W_{hh}$ are weight matrices, $b_{h}$ is the bias vector for the hidden layer, $W_{yh}$ is the weight matrix and $b_{y}$ is the bias vector for the output layer. *f* and *g* are activation functions, which are often the hyperbolic tangent or the rectified linear unit (ReLU) function.

#### 2.2.2. Gated Recurrent Unit (GRU) [39]

Gated Recurrent Units (GRUs) are a type of Recurrent Neural Network (RNN) designed to capture the long-term dependencies between time steps in sequential data. Unlike traditional RNNs, which have a simple linear activation function to capture the relationships between time steps, GRUs have a gating mechanism that allows them to selectively choose which information to preserve from previous time steps and which to discard.

The GRU has two hidden states, the reset gate and the update gate, which are used to control the flow of information from previous time steps. The reset gate decides how much of the previous hidden state is to be forgotten, while the update gate decides how much of the previous hidden state is to be combined with the current input. The final hidden state is then used to predict the output at the current time step.

The GRU’s gating mechanism allows it to efficiently handle long sequences of data, as it can selectively preserve the most relevant information and discard the rest. Additionally, GRUs require fewer parameters than traditional RNNs, which can reduce overfitting and improve model training efficiency.

GRUs have been applied to various time series forecasting problems, including solar PV output forecasting. In such applications, the GRU is trained on historical time series data to capture the relationships between time steps and then used to make predictions for future time steps. The input to the GRU is the time series data, and the output is a prediction for the future values of the time series. The architecture of a single block of GRU is shown in Figure 4.

The update gate, reset gate and new memory cell vector of a Gated Recurrent Unit (GRU) can be computed using the following equations:(3)$$z_{t}=\sigma \left(W_{z}x_{t}+U_{z}h_{t-1}+b_{z}\right)$$
(4)$$r_{t}=\sigma \left(W_{r}x_{t}+U_{r}h_{t-1}+b_{r}\right)$$
(5)$$\tilde{h_{t}}=\tanh\left(Wx_{t}+U\left(r_{t}⊙h_{t-1}\right)+b\right)\;$$
where $x_{t}$ is the input at time step *t*; $h_{t-1}$ is the hidden state at the previous time step; $W_{z},U_{z}$ and $b_{z}$ are the weights and bias for the update gate; $W_{r},U_{r}$ and $b_{r}$ are the weights and bias for the reset gate; W,U and *b* are the weights and bias for computing the new memory cell vector; $z_{t}$ is the update gate output; $r_{t}$ is the reset gate output; $\tilde{h_{t}}$ is the candidate hidden state; and $h_{t}$ is the current hidden state. The σ function is the sigmoid function, and ⊙ is the element-wise product (Hadamard product).

#### 2.2.3. Long Short-Term Memory (LSTM) [37]

Long Short-Term Memory (LSTM) is a type of Recurrent Neural Network (RNN) architecture that is specifically designed to overcome the vanishing gradient problem faced by traditional RNNs. LSTM is widely used for time-series analysis, sequential prediction and natural language processing tasks.

LSTMs consist of memory cells that store information for an extended period of time and gates that control the flow of information into and out of the memory cells. The three gates in LSTM are the input gate, forget gate and output gate. These gates help in deciding what information should be stored in the memory, what information should be discarded and what information should be outputted.

The input gate controls the amount of new information that is allowed to enter the memory cell. The forget gate decides what information should be discarded from the memory cell. The output gate controls what information should be outputted from the memory cell.

LSTMs are trained using backpropagation through time (BPTT) to minimize a loss function that represents the difference between the predicted output and the true output. The weights of the gates and the memory cells are updated during the training process to minimize the loss function.

LSTMs are able to capture long-term dependencies in time-series data and to outperform traditional RNNs in tasks that require memory and sequential prediction. They have been widely used for time-series forecasting, natural language processing and speech recognition. The architecture of a single block of LSTM is shown in Figure 5.

The equations for the different gates of LSTM are as follows:(6)$$f_{t}=\sigma \left(W_{f}\left[h_{t-1},x_{t}\right]+b_{f}\right)$$
(7)$$i_{t}=\sigma \left(W_{i}\left[h_{t-1},x_{t}\right]+b_{i}\right)$$
(8)$$\tilde{C_{t}}=\tanh\left(W_{C}\left[h_{t-1},x_{t}\right]+b_{C}\right)$$
(9)$$C_{t}=f_{t}*C_{t-1}+i_{t}*\tilde{C_{t}}$$
(10)$$o_{t}=\sigma \left(W_{o}\left[h_{t-1},x_{t}\right]+b_{o}\right)$$
(11)$$h_{t}=o_{t}*\tanh\left(C_{t}\right)$$
where $f_{t}$ is the forget gate, $i_{t}$ is the input gate, $\tilde{C_{t}}$ is the candidate cell state, $C_{t}$ is the cell state, $o_{t}$ is the output gate, $h_{t}$ is the hidden state at time *t*, $x_{t}$ is the input at time *t*, *W* and *b* are the weight and bias matrices, and σ is the sigmoid activation function.

#### 2.2.4. Transformer [24]

Transformer is a neural network architecture that was introduced in 2017 by Vaswani et al. in the paper “Attention is All You Need” [24]. Transformer is designed to handle sequential data and has revolutionized the field of natural language processing (NLP). The key innovation of Transformer is the self-attention mechanism, which allows the model to weigh the importance of each feature in the input sequence when making predictions. The original transformer and its variants have also been used in multiple time series forecasting applications [24,40].

A traditional Recurrent Neural Network (RNN) operates on a sequence by processing one element at a time while maintaining an internal state. In contrast, the Transformer operates on the entire sequence at once, allowing it to capture long-term dependencies between elements. This capability of Transformer makes it well suited for sequence-to-sequence problems, such as time series forecasting.

The Transformer architecture consists of an encoder and a decoder, both of which are made up of a series of stacked attention and feed-forward layers. The encoder takes the input sequence and computes a sequence of hidden states. The decoder then takes the hidden states and produces the output sequence.

The attention mechanism in Transformer computes a weight for each element in the input sequence, indicating its importance in the current prediction. These weights are used to compute a weighted sum of the hidden states, which is then used to make the prediction. This allows Transformer to focus on the most relevant parts of the input sequence when making predictions.

In addition to the self-attention mechanism, Transformer also uses multi-head attention, which allows the model to attend to multiple aspects of the input sequence at once. This allows the model to capture complex relationships between elements in the input sequence. Figure 6 shows the architecture of a transformer.

The self-attention in the Transformer architecture is given by the following equation:(12)$$\mathrm{Attention}\left(Q,K,V\right)=\mathrm{softmax}\left(\frac{QK^{T}}{\sqrt{d_{k}}}\right)V$$

This equation calculates the output of the self-attention mechanism, which is used to compute the relationship between each position in the input sequence. Here, *Q* represents the queries, *K* represents the keys and *V* represents the values. The equation calculates the dot product of the queries and keys, which are divided by the square root of the dimensionality of the keys, to scale the gradients. The result is passed through a softmax activation function to obtain a probability distribution over the keys, which is then used to weight the values.

The multi-head attention in the transformer architecture is given by the following equation:(13)$$\mathrm{MultiHead}\left(Q,K,V\right)=\mathrm{Concat}\left({\mathrm{head}}_{1},...,{\mathrm{head}}_{h}\right)W^{O}$$
where
$${\mathrm{head}}_{i}=\mathrm{Attention}\left(QW_{i}^{Q},KW_{i}^{K},VW_{i}^{V}\right)$$

This equation applies the self-attention mechanism multiple times in parallel to capture different relationships between the input sequence elements. Here, *Q*, *K* and *V* are the same as in the self-attention equation but are projected using weight matrices $W_{i}^{Q}$, $W_{i}^{K}$ and $W_{i}^{V}$ to create *h* different subspaces, or "heads". The self-attention operation is then applied to each of these subspaces to produce *h* different outputs, which are concatenated and linearly transformed using a weight matrix $W^{O}$ to produce the final output.

### 2.3. Forecast Settings

The forecasts were performed in three different settings, namely multi-in multi-out, multi-in uni-out and uni-in uni-out. For all three of these settings, the lookback length (LBL) is the number of past data points used as input and forecast horizon length (FHL) is the number of datapoints into the future that forecasts was made. Multi-in muli-out is the scenario where, for fixed values of LBL and FHL, the forecasts for multiple sites are made using the input past data from each of those sites. This is useful when multiple site are to be monitored and managed simultaneously. Similarly, in the multi-in uni-out setting for fixed values of LBL and FHL, the forecasts for a single site are made using past input data from multiple neighbouring sites. The multi-in uni-out setting is useful if the goal is to obtain longer term forecasts of a single site with more accuracy. Finally, in the uni-in uni-out setting for fixed values of LBL and FHL, the forecasts for a single site are made using past input data from the same site. This setting is useful when only single-site short-term forecasts are required and resources are not available for multi-in uni-out settings. These resources might be computational resources or data. In terms of computational resources, processing multiple site data points requires more computation as compared with processing single-site data points. In terms of data, it is possible that only a single PV site is in operation or that the user may not have access to other neighbouring sites data. These three settings have been visualized in Figure 7.

## 3. Experiment and Results

The possibility of forecasting solely using only the historical recordings of the solar PV output in the absence of any kind of weather information was tested in these experiments. Famous deep learning architectures—RNN, GRU, LSTM and Transformer—were compared here in terms of mean square error (MSE) and mean absolute error (MAE) while making forecasts.

### 3.1. Implementation Details

The data were split into train, test and validation sets at a ratio of 8:1:1. All of the models were then trained with L2 loss, which is the squared difference between the actual value and the prediction value, using the ADAM [41] optimizer. The code used was implemented using the PyTorch framework, and the the experiments were performed on a single NVIDIA GeForce RTX 3060 laptop GPU.

Furthermore, the experiments were performed in three different setups: (i) the multi-in multi-out setting, (ii) the multi-in uni-out setting and (iii) the uni-in uni-out setting.

### 3.2. Details of Hyper-Parameters Used

The details of the hyper-parameters used are shown in Table 2. The hyper-parameter values are same for RNN, GRU and LSTM for all settings. The learning rate was set very low to 1 × 10^−5^ as the models seemed to overfit even in very few epochs when set to a value higher than this. Additionally, due to the same reason, each configuration of the models used were also very simple. The number of RNN, GRU and LSTM layers used was 2 and the number of encoders and decoders in Transformer was set to 3. The weight decay was set to 1 × 10^−6^, and the batch size was set to a fixed value of 64 for each of the settings. The number of epochs for Transformer used was 50 as compared with 100 for the rest of the models because Transformer was overfitting very early.

### 3.3. Evaluation Metrics

The evaluation metrics used to evaluate the performance of the forecasting models are mean square error (MSE) and mean absolute error (MAE).

#### 3.3.1. Mean Square Error (MSE)

MSE is a common metric used to evaluate the accuracy of a regression model. It measures the average squared difference between the predicted values and the actual values. The formula for calculating MSE is as follows:(14)$$MSE=\frac{1}{n}\sum {\left(y-\hat{y}\right)}^{2}$$
where *n* is the number of observations, *y* is the actual value and *ŷ* is the predicted value. Better performance is obtained during forecasting when the MSE is lower. It amplifies the effect of a larger error, and the model will be more sensitive to them.

#### 3.3.2. Mean Absolute Error (MAE)

MAE is another metric commonly used to evaluate regression models. It measures the average absolute difference between the predicted values and the actual values. The formula for calculating MAE is as follows:(15)$$MAE=\frac{1}{n}\sum |y-\hat{y}|$$
where *n* is the number of observations, *y* is the actual value and *ŷ* is the predicted value. MAE has the same unit as the predicted output and is more interpretable. Additionally, the performance of the model is better when the MAE value is lower. MAE is less sensitive to outliers and is easy to understand.

### 3.4. Results

#### 3.4.1. Results of Multi-In Multi-Out Setting

The comparative results obtained for the multi-in multi-out setting for each of the used forecasting models are shown in Table 3. From Table 3, it can be deduced that Transformer performs arguably the best in the multivariate predictive scenario. This can be attributed to the better ability of Transformer to realize the individual relationships between each of the input features. As can be seen from Table 3, Transformer outperforms each of the other architectures by a huge margin. The RNN, GRU and LSTM models perform almost similarly when the length of the lookback window is 48 or 96, with either GRU or LSTM mostly performing better than the simple RNN model. The performance of LSTM is much better as compared with that of RNN and GRU when the lookback window length was 144 and the forecast horizon was 24. Furthermore, while for a short FHL length of 4 or 8, the performances are comparable for all LBLs, it can even be said that the performance when a shorter LBL, 48, is used is better. However, as the FHL becomes longer (12 or 24), the performance improves as the FHL increases. Therefore, Table 3 suggests that, when proper resources are available, the use of Transformer for the multivariate setting would provide the best results. However, architectures such as LSTM and GRU can also be used if required, since the forecasting results are acceptable in these cases as well. Additionally, for a shorter FHL, using a shorter LBL is sufficient; however, for a longer FHL, a longer LBL is required.

#### 3.4.2. Results of Multi-In Uni-Out Setting

Table 4 gives the comparative results for the forecasting models in the multi-in uni-out setting. Table 4 shows that the LSTM and Transformer models go head to head in terms of different settings. The LSTM has better MAE values in most of the cases in this setting and Transformer has better MSE values with all the lookback window lengths and forecast horizon lengths. However, in the multi-in multi-out setting, there is little difference in the performance of the best and the worst performing models in each of the settings. Additionally, like in multi-in multi-out for a short FHL of 4 or 8, the performances are comparable for all LBLs, and it can even be said that, when a shorter LBL of 48 is used, the performance is better. However, as the FHL becomes longer (12 or 24), the performance improves as the FHL increases. Therefore, in this scenario, it can be suggested that RNN, GRU or LSTM can be used, considering the fact that there is no major difference in performance in each of the four models and the fact that Transformer requires much larger computing resources as compared with rest of the models. Furthermore, like in the multi-in multi-out setting with a shorter FHL, using a shorter LBL is sufficient; however, for a longer FHL, a longer LBL is required.

#### 3.4.3. Results of Uni-In Uni-Out Setting

The results of the uni-in uni-out forecasting of the power of a solar pv is shown in Table 5. In this setting, Transformer mostly dominates the other models in terms of performance. The rest of the models performed better than the Transformer model only occasionally. For example, RNN performed better than the rest of the models in terms of mean squared error when the lookback length was 48 and the forecast horizon was 4. Similarly, GRU performed better than the rest of the models in terms of mean squared error when the lookback length was 48 and the horizon was 12 and when the lookback length was 96 and the horizon was 4. Finally, LSTM performed better than the rest of the models both in terms of mean squared error and mean average error when the lookback length was 48 and the forecast horizon was 8 and only in terms of mean squared error when the lookback length was 96 and the forecast horizon was 4. Like in the previous two settings for a shorter FHL, the performances are comparable for different LBLs. For RNN, the performance for an FHL of 4 was even better when the LBL was 48 compared with an LBL of 96 or 144. However, for a longer FHL to be forecasted, the results are better when a longer LBL is used.

#### 3.4.4. Comparative Analysis of Results Obtained

The three forecast settings: multi-in multi-out, multi-in uni-out and uni-in uni-out are useful in their own regards. A visual comparison of Table 3, Table 4 and Table 5 shows that these settings play different roles when the FHLs are varied for each LBL and each model. The simplest of the three settings, the uni-in uni-out setting, seems to be much more effective than the rest when the required FHL is short (4 or 8). Similarly, the multi-in uni-out setting seems to be more effective when the FHLs are longer (12 or 24). So, either the multi-in uni-out or uni-in uni-out setting can be used based on the FHL required, when the forecast of only a single site is required. Finally the results of mutli-in multi-out are better than that of the multi-in uni-out setting when the FHL is 4 or 8 but not as good as the uni-in uni-out setting for the same FHL. For a longer FHL the results of the multi-in multi-out setting are not as good as that of the other two. However, the multi-in multi-out setting is still useful when the forecasts of all the sites are required at once without having to generate the forecast of individual sites separately.

The plots for the output by each of the used forecasting models are presented in Figure 8. The plots in Figure 8 show that the forecast does not always follow the exact patterns of the actual output. However, the forecasts are actually really close to the real output values. Moreover, the plots for the Transformer model seem to model the output patterns better than the rest of the models.

## 4. Conclusions

Unlike previously performed studies on solar PV output forecasting, this study has opted to perform forecasts solely based on historical reading of the PV output values and without considering weather information. This study has been performed so as to confirm whether the forecast of solar PV output values in such conditions is suitable. The data used were collected from multiple solar PV sites in Suncheon, South Korea, at regular intervals for a duration of 6 months. The forecasting models used for the study are RNN, GRU, LSTM and Transformer. This study was performed in three different settings (multivariate, multi-in uni-out and univariate). For each of the different settings, Transformer seemed to have performed better most of the time. However, it is necessary to consider that the Transformer model requires more computing resources as compared to the rest of the models. Therefore, it is best that a proper forecasting model is selected based on the requirements and existing constraints such as the availability of resources and data. For instance, the results for the multi-in uni-out setting were not very different for each of the models, so if there a possible resource constraint, then the less powerful models such as LSTM, GRU or even RNN can be considered. Furthermore, it is necessary that the data in use are well processed and that there are very few anomalies in them. If these steps are properly performed, then the forecasting requirements of solar PV power plant outputs can be achieved even in the absence of meteorological data.

In future work, various existing architectures can be studied to perform solar output forecasts and a novel architecture can be suggested. Longer term forecasts can also be studied for detailed system implementation for load management. Finally, the forecasted results can also be used to detect anomalies in the system by comparing the actual output of the plants with forecasted outputs.

## Figures and Tables

**Figure 1 sensors-23-03399-f001:**
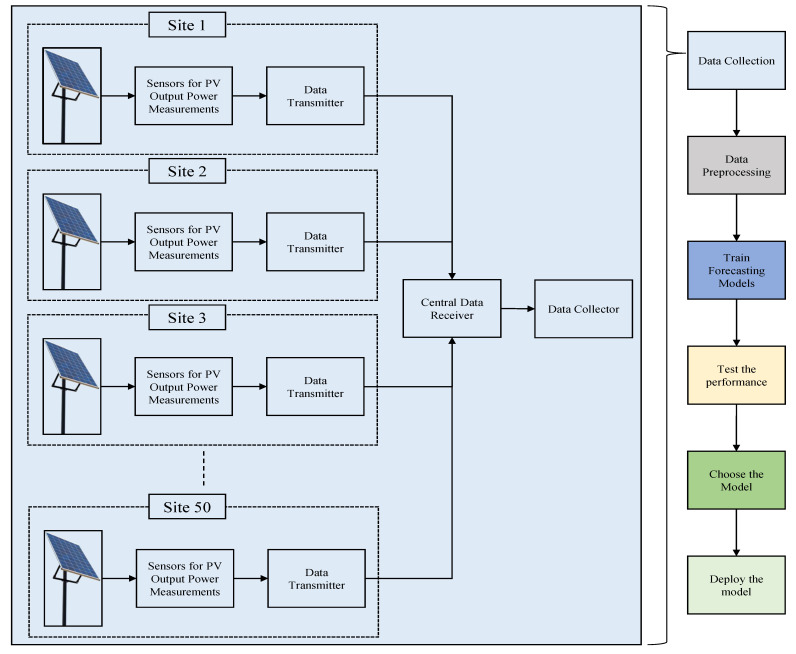
Flowchart of the overall steps performed.

**Figure 2 sensors-23-03399-f002:**
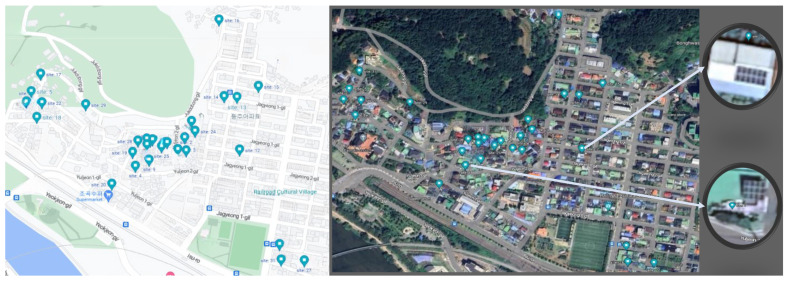
View of the locations of some of the PV sites used in this study.

**Figure 3 sensors-23-03399-f003:**

The architecture of a simple RNN unrolled.

**Figure 4 sensors-23-03399-f004:**
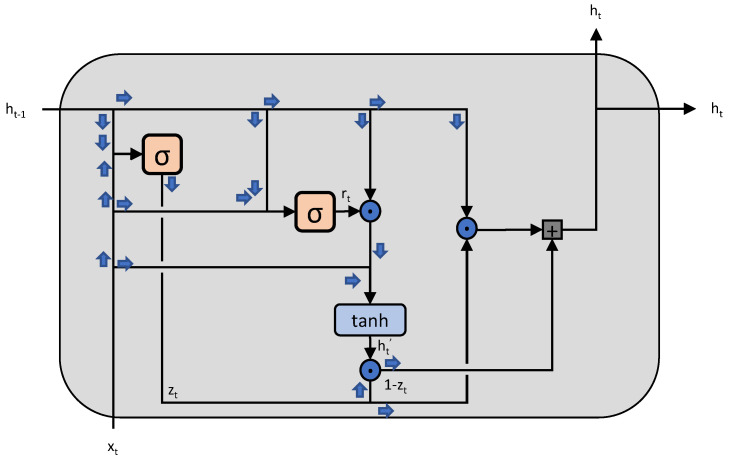
The architecture of a single block of GRU.

**Figure 5 sensors-23-03399-f005:**
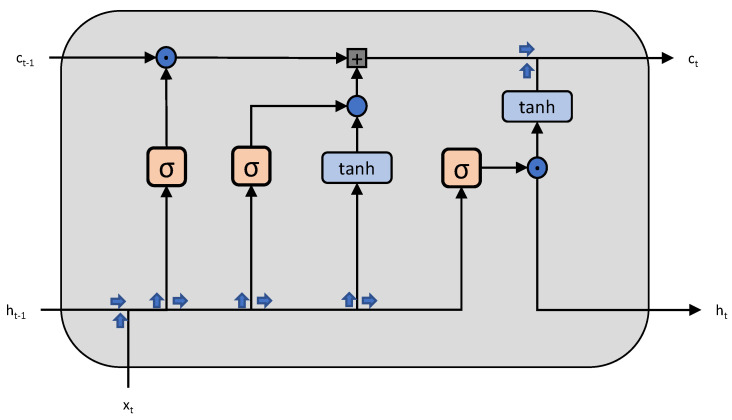
The architecture of a single block of LSTM.

**Figure 6 sensors-23-03399-f006:**
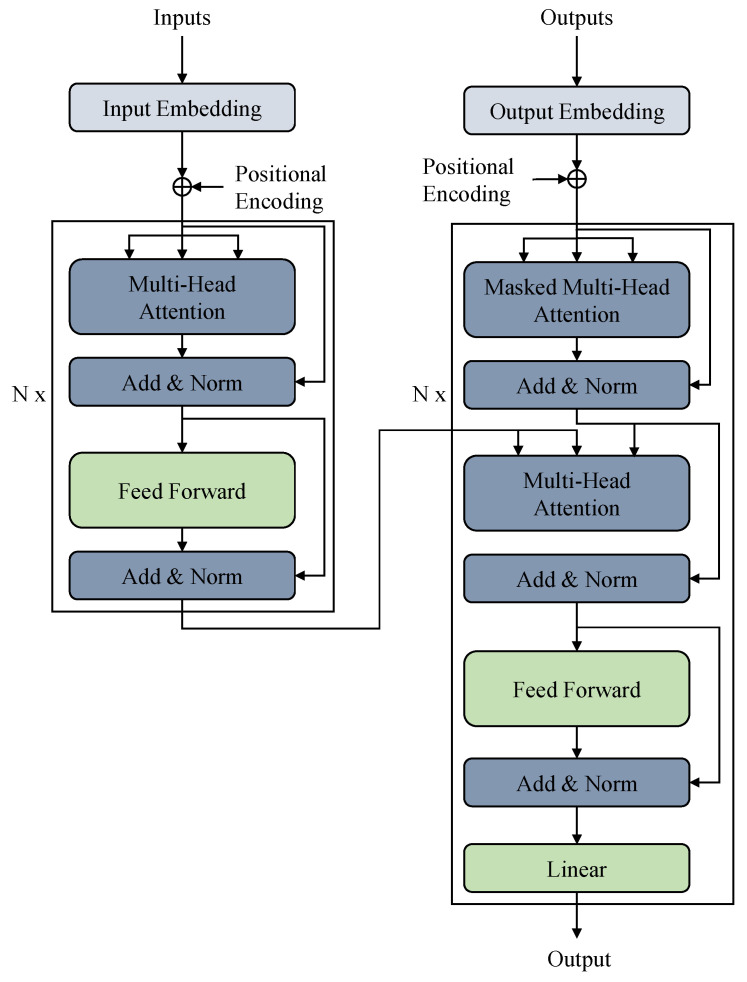
The architecture of Transformer [24].

**Figure 7 sensors-23-03399-f007:**
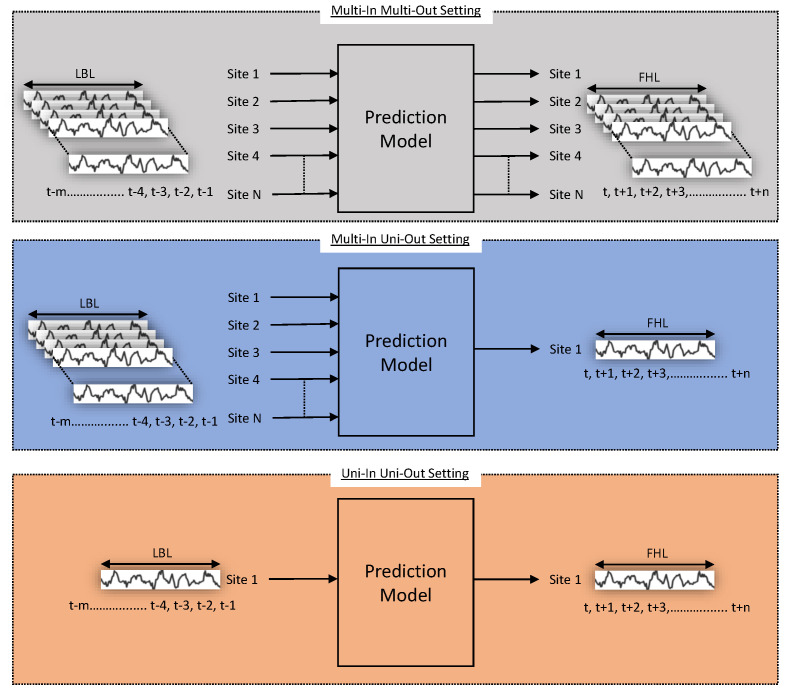
Visualization of forecast settings.

**Figure 8 sensors-23-03399-f008:**
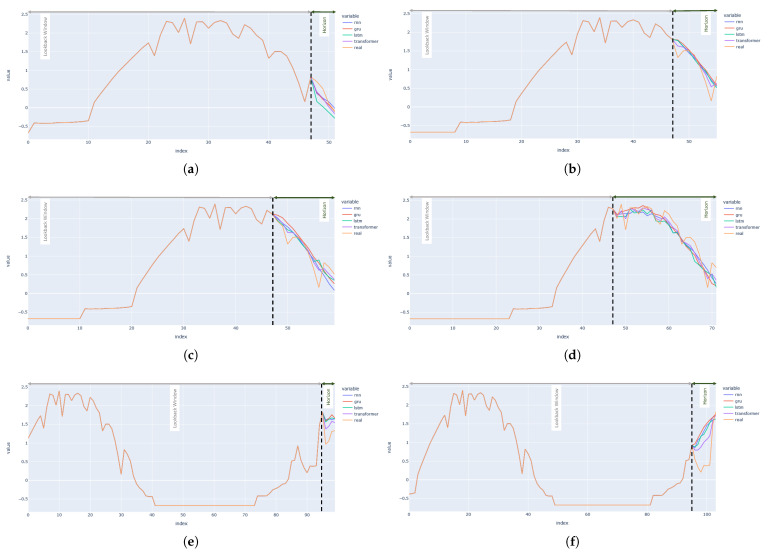
Visualization of the forecasts of PV output from RNN, GRU, LSTM and Transformer in the univariate setting for varying lookback lengths and forecast horizons. Lookback length has been written as ’LB’, and forecast horizon has been written as ’H’. (**a**) LB:48 H:4; (**b**) LB:48 H:8; (**c**) LB:48 H:12; (**d**) LB:48 H:24; (**e**) LB:96 H:4; (**f**) LB:96 H:8; (**g**) LB:96 H:12; (**h**) LB:96 H:24; (**i**) LB:144 H:4; (**j**) LB:144 H:8; (**k**) LB:144 H:12; (**l**) LB:144 H:24.

**Table 1 sensors-23-03399-t001:** A small portion of the dataset used.

Time	Site 1	Site 2	Site 3	Site 4	Site 5	.......	Site 47	Site 48	Site 49	Site 50
2020-01-01 06:45	0	0	0	0	0	.......	0	0	0	0
2020-01-01 07:00	0	0	0	0	0	.......	0	0	0	0
2020-01-01 07:15	0	0	0	0	0	.......	0	0	0	0
2020-01-01 07:30	0	207	216	0	225	.......	0	186	217	215
2020-01-01 07:45	212	205	217	217	218	.......	212	192	265	215
2020-01-01 08:00	211	250	212	271	211	.......	212	494	465	215
2020-01-01 08:15	225	377	209	363	585	.......	214	745	708	235
2020-01-01 08:30	240	424	865	648	798	.......	239	953	934	250
2020-01-01 08:45	260	541	1087	948	1017	.......	321	1138	1147	306
2020-01-01 09:00	506	861	1278	1147	1251	.......	428	1315	1322	505
.......	.......	.......	.......	.......	.......	.......	.......	.......	.......	.......
2020-06-22 14:15	794	1176	1738	1177	715	.......	1447	1349	1370	1417
2020-06-22 14:30	839	885	970	866	972	.......	792	892	897	780
2020-06-22 14:45	911	1043	1021	1093	458	.......	1489	1211	1241	1419
2020-06-22 15:00	1681	643	1130	659	591	.......	567	693	703	567
2020-06-22 15:15	1474	1032	1123	823	275	.......	1086	1057	947	1007
.......	.......	.......	.......	.......	.......	.......	.......	.......	.......	.......

**Table 2 sensors-23-03399-t002:** Details of hyper-parameters used.

RNN, LSTM, GRU		Transformer	
Hyper-Parameter	Value	Hyper-Parameter	Value
Number of hidden state	64	Number of heads	4
Number of recurrent layers	2	Number of encoder layers	3
		Number of decoder layers	3
		Number of expected features in the encoder/decoder inputs	128
		Feedforward network dimension	256
Number of epochs	100	Number of epochs	50
Dropout	0.3	Dropout	0.3
Weight Decay	$1\times 10^{-6}$	Weight Decay	$1\times 10^{-6}$
Learning rate	$1\times 10^{-4}$	Learning rate	$1\times 10^{-4}$

**Table 3 sensors-23-03399-t003:** Results in multi-in multi-out setting. Lower MSE and MAE values are better. The best results are shown in bold and underlined.

	Models	RNN		GRU		LSTM		Transformer	
LBL	FHL	MSE	MAE	MSE	MAE	MSE	MAE	MSE	MAE
48	4	0.1337	0.2111	0.1280	0.2049	0.1334	0.2084	**0.1070**	**0.1786**
	8	0.2351	0.3103	0.1643	0.2479	0.1927	0.252	**0.1429**	**0.2245**
	12	0.2059	0.2796	0.2414	0.3027	0.2197	0.2739	**0.1856**	**0.2618**
	24	0.2514	0.3168	0.237	0.2946	0.2998	0.3199	**0.2352**	**0.2768**
96	4	0.1356	0.2109	0.1345	0.2081	0.1377	0.2052	**0.0971**	**0.1722**
	8	0.1657	0.2409	0.1574	0.2387	0.1612	0.2286	**0.1137**	**0.2113**
	12	0.2386	0.2927	0.2165	0.2828	0.2646	0.2881	**0.1603**	**0.2404**
	24	0.276	0.3416	0.2709	0.3113	0.4467	0.4007	**0.2069**	**0.2451**
144	4	0.1472	0.2203	0.1427	0.2174	0.1385	0.2088	**0.115**	**0.1823**
	8	0.2331	0.2985	0.158	0.2336	0.2523	0.1887	**0.1246**	**0.2345**
	12	0.3148	0.3478	0.2233	0.2851	0.2465	0.1773	**0.1687**	**0.2312**
	24	0.3475	0.3800	0.4703	0.3966	0.2359	0.2997	**0.184**	**0.2211**

**Table 4 sensors-23-03399-t004:** Results in multiple-in uni-out setting. Lower MSE and MAE values are better. The best results are shown in bold and underlined.

	Models	RNN		GRU		LSTM		Transformer	
LBL	FHL	MSE	MAE	MSE	MAE	MSE	MAE	MSE	MAE
48	4	0.1366	0.2249	0.1435	0.2264	**0.1112**	0.1995	0.1120	**0.1762**
	8	0.1382	0.2265	0.1382	0.228	**0.1157**	0.2083	0.1184	**0.1834**
	12	0.1454	0.2488	0.1369	0.2264	**0.1104**	0.2049	0.1212	**0.1893**
	24	0.2217	0.3054	0.1717	0.2594	0.1517	0.2405	**0.1300**	**0.2020**
96	4	0.1335	0.2249	0.1446	0.2259	**0.1030**	0.1947	0.1171	**0.1763**
	8	0.1474	0.2331	0.133	0.2199	**0.1154**	0.1980	0.1245	**0.1923**
	12	0.1410	0.2417	0.1301	0.2161	**0.1087**	0.1975	0.1217	**0.1838**
	24	0.2178	0.2913	0.2174	0.2901	0.1919	0.2707	**0.1747**	**0.2216**
144	4	0.1396	0.2281	0.1494	0.2304	0.1106	0.1978	**0.0993**	**0.1624**
	8	0.147	0.2278	0.1399	0.2254	0.1237	0.2076	**0.1153**	**0.1669**
	12	0.1472	0.244	0.1347	0.2256	**0.1054**	0.1923	0.1192	**0.1823**
	24	0.2457	0.3220	0.2096	0.2812	0.1801	0.2550	**0.1562**	**0.2112**

**Table 5 sensors-23-03399-t005:** Results in uni-in uni-out setting. Lower MSE and MAE values are better. The best results are shown in bold and underlined.

	Models	RNN		GRU		LSTM		Transformer	
LBL	FHL	MSE	MAE	MSE	MAE	MSE	MAE	MSE	MAE
48	4	**0.0887**	0.1560	0.0917	0.1624	0.0923	0.1621	0.094	**0.1518**
	8	0.1140	0.2099	0.1066	0.1890	**0.1054**	**0.1815**	0.1128	0.1914
	12	0.1248	0.2241	**0.1204**	0.2033	0.1238	0.2094	0.1247	**0.1924**
	24	0.201	0.2811	0.1947	0.2671	0.2167	0.2788	**0.1832**	**0.2531**
96	4	0.0892	0.1616	0.0900	0.1543	**0.0879**	0.1577	0.0912	**0.1423**
	8	0.1057	0.1963	0.1031	0.1765	0.1004	0.1755	**0.0996**	**0.1724**
	12	0.1272	0.2197	0.1276	0.1989	0.1202	0.2056	**0.1065**	**0.1818**
	24	0.1968	0.2737	0.2324	0.2865	0.2055	0.2723	**0.1624**	**0.2395**
144	4	0.0949	0.1628	0.0927	0.1577	0.0931	0.161	**0.0832**	**0.1463**
	8	0.1087	0.1925	0.1096	0.1837	0.1049	0.1828	**0.0999**	**0.1582**
	12	0.132	0.2281	0.1266	0.2027	0.1216	0.2037	**0.1123**	**0.1812**
	24	0.2341	0.2973	0.2414	0.2927	0.221	0.2848	**0.2120**	**0.2541**

## Data Availability

Not applicable.

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
