# Peer review of "Predicting the Output of Solar Photovoltaic Panels in the Absence of Weather Data Using Only the Power Output of the Neighbouring Sites"

_sensors, 2023, doi:10.3390/s23073399_

Round 1

Reviewer 1 Report

The paper proposes to verify the effectiveness of several deep learning methods on the predicting of the output of solar photovoltaic panels, with just the recorded values. The transformer method outperforms RNN, GRU and LSTM in the collected dataset. There are some serious issues that need to be properly solved.

1) The novelty of the paper is limited, because only several deep learning methods are used as baselines to verify the prediction performance. So, what is the contribution of the model?

2) In section 3, examples of the data should be given, otherwise, we have no idea what the data is.

3) Please give a figure to explain the three settings, i.e., multivariate setting, multi-in single-out setting and univariate setting. Why use different settings, and are there any practical applications?

4) Explain the LBL and FHL, because the readers may not understand the experiments.

5) Some visualizing Experiments are required to further show and compared the performance of different methods.

6) Analysis of different methods under different settings need to be given.

7) The performance is related to the hyper-parameters, so I am not sure the comparison is fair because different models have distinct parameter sizes. Please show this.

8) A conclusion is usually required.

Author Response

Reply to Reviewer 1

I would like to thank the reviewer for spending their valuable time reviewing my manuscript. My responses to the comments are as follows:

Comment 1: The novelty of the paper is limited, because only several deep learning methods are used as baselines to verify the prediction performance. So, what is the contribution of the model?  

Response: Most of the previous works exploring the field of solar PV output forecasting make use of meteorological data to make the forecasts. However, due to several issues (faulty sensors or equipment, power outages etc.) there may be instances that the weather data are not always available. Considering such scenario, I have studied the possibility of utilizing just the output readings of the solar PV for making the forecasts. Rather than suggesting a novel architecture for forecasting, I have performed extensive experimentation with the existing machine learning architectures on different settings to study the suitability of using just the PV output data for forecasting. Furthermore, via experimentation I have suggested what kind of architecture is more suitable for each of settings.

Comment 2: In section 3, examples of the data should be given, otherwise, we have no idea what the data is.

Response: The example of the data has now been added as asked. However, it has been added to section 2.1 in the data description section.

Comment 3: Please give a figure to explain the three settings, i.e., multivariate setting, multi-in single-out setting and univariate setting. Why use different settings, and are there any practical applications?

Response: Explanation of the three settings have been added in section 2.3. Additionally, to avoid confusion, the settings have been renamed as i.e., multi-in multi-out setting, multi-in single-out setting and uni-in uni-out setting.

The different settings can be utilized for different use cases based on the resources available and the requirements. Multivariate setting: when multiple site are to be monitored and managed simultaneously. Multi-in single-out: is useful if the goal is obtaining longer term forecasts of a single site with more accuracy. Univariate: When only the single site short term forecasts are required and the resources are not available for multi-in uni-out settings. These resources might be compute resources or data. In terms of compute resources, processing multiple site data points require more compute as compared to processing single site data points. In terms of data, it is possible that only single PV site is in operation or the user may not have access to other neighboring sites data.

Comment 4: Some visualizing Experiments are required to further show and compared the performance of different methods.

Response: The output visualization graphs are present at the end.

Comment 5: Analysis of different methods under different settings need to be given.

Response: Further analysis have been added to the results section 3.3.4.

Comment 6: The performance is related to the hyper-parameters, so I am not sure the comparison is fair because different models have distinct parameter sizes. Please show this.

Response: The details of the hyper-parameter settings have been added as asked.

Comment 7: A conclusion is usually required.

Response: Conclusion section has been added asked.

Reviewer 2 Report

I think the paper is suitable for publication in the journal after satisfying the following issues:

A large number of citations are necessary in lines 46, 47 and 48. The author indicates that there are numerous machine learning models but he/she does not cite apropriately.

Indicate your contribution, showing the contributions of previous works and highlighting the novelty of the research work. The author must write a new paragraph in the introduction citing previous authors and making the corresponding analysis.

Please improve grammar of the last paragraph of introduction, lines 60 to 64

Please move Figure 1 after be mentioned in the text

The author uses the pronoun we in several parts of the document, however I can see that there is only one author signing the document. please modified apropriately.

Please correct the use of the determiner "the", in several parts of the document it is used inappropriately. The section, the Equation, the Table, etc.

The results section needs to be rewritten again to properly display the findings. The author frequently uses the phrase "...perform better...", however, he/she does not specify within the calculated metrics what it means that the algorithm performs better. I recommend for each figure and for each table a detailed description, and then analysis of their corresponding data.

I would change the title of the discussion section to conclusion.

Please place figure 7 after the first citation in the text.

Author Response

I would like to thank the reviewer for spending their valuable time reviewing my manuscript. My responses to the comments are as follows:

Comment 1: A large number of citations are necessary in lines 46, 47 and 48. The author indicates that there are numerous machine learning models but he/she does not cite apropriately.

Response: Further citations have been added to the mentioned lines.

Comment 2:  Indicate your contribution, showing the contributions of previous works and highlighting the novelty of the research work. The author must write a new paragraph in the introduction citing previous authors and making the corresponding analysis.

Response: Additional content indicating the contribution of this manuscript has been added to the introduction part along with the analysis of the previous works.

Comment 3: Please improve grammar of the last paragraph of introduction, lines 60 to 64.

Response: The change has been made as asked by the reviewer.

Comment 4: Please move Figure 1 after be mentioned in the text.

Response: The Figure 1 has been moved after it’s reference in the text.

Comment 5: The author uses the pronoun we in several parts of the document, however I can see that there is only one author signing the document. please modified apropriately.

Response: The manuscript has been changed accordingly.

Comment 6: Please correct the use of the determiner "the", in several parts of the document it is used inappropriately. The section, the Equation, the Table, etc.

Response: The changes have been made accordingly.

Comment 7: The results section needs to be rewritten again to properly display the findings. The author frequently uses the phrase "...perform better...", however, he/she does not specify within the calculated metrics what it means that the algorithm performs better. I recommend for each figure and for each table a detailed description, and then analysis of their corresponding data.

Response: The results section has been rewritten with a more detailed analysis.

Comment 8: I would change the title of the discussion section to conclusion.

Response: The title of the discussion has been changed to conclusion as asked.

Comment 9: Please place figure 7 after the first citation in the text.

Response: The Figure 7 has been moved after it’s reference in the text.

Reviewer 3 Report

The concept of the article is good. Kindly check the following comments to enhance your article. 

1. The abstract is not well-articulated. The technical objective should be mentioned and rewritten the abstract with concise information. 

2. The introduction section should be framed by combining the sections 0 and 1. Some of the basic information can be removed from the introduction. 

3. Please avoid lumped references. Few places, lumped references were added. Authors must explain each reference along with their limitations/drawbacks. 

4. The authors contribution should be pointed out at the end of the introduction section. 

5. The concept of the article is good, however, authors felt the innovation/novelty is weak in the current version. Authors must justify with your answers. 

6. What are the reasons for taking only power data alone? 

7. The forecasting models used in the article are well-known? What is the new information in this article?

8. Conclusion must be added separately. change the heading "Discussion" to "conclusion"

Author Response

I would like to thank the reviewer for spending their valuable time reviewing my manuscript. My responses to the comments are as follows:

Comment 1: The abstract is not well-articulated. The technical objective should be mentioned and rewritten the abstract with concise information.

Response: The abstract has been re-written with further details on the technical objectives and more concise information.

Comment 2:  The introduction section should be framed by combining the sections 0 and 1. Some of the basic information can be removed from the introduction.

Response: The introduction and related works have been combined.

Comment 3: Please avoid lumped references. Few places, lumped references were added. Authors must explain each reference along with their limitations/drawbacks.

Response: The lumped references have only been used wherever a common point is presented by each of the referred papers.

Comment 4: The authors contribution should be pointed out at the end of the introduction section.

Response: The contributions have been added at the end of the introduction.

Comment 5: The concept of the article is good, however, authors felt the innovation/novelty is weak in the current version. Authors must justify with your answers.

Response: Most of the previous works exploring the field of solar PV output forecasting make use of meteorological data to make the forecasts. However, due to several issues (faulty sensors or equipment, power outages etc.) there may be instances that the weather data are not always available. Considering such scenario, I have studied the possibility of utilizing just the output readings of the solar PV for making the forecasts. Rather than suggesting a novel architecture for forecasting, I have performed extensive experimentation with the existing machine learning architectures on different settings to study the suitability of using just the PV output data for forecasting. Furthermore, via experimentation I have suggested what kind of architecture is more suitable for each of settings.

Comment 6: What are the reasons for taking only power data alone?

Response: As discussed in the response of previous comment, most of the previous works exploring the field of solar PV output forecasting make use of meteorological data to make the forecasts. However, due to several issues (faulty sensors or equipment, power outages etc.) there may be instances that the weather data are not always available. Considering such scenario, I have studied the possibility of utilizing just the output readings of the solar PV for making the forecasts.

Comment 7: The forecasting models used in the article are well-known? What is the new information in this article?

Response: This manuscript presents a study to understand the suitability of utilizing just the solar PV output data for the forecast. This includes how well the output can be forecasted utilizing the already existing machine learning models. I have discussed how well each of the explored models perform in different scenarios.

Comment 8: Conclusion must be added separately. change the heading "Discussion" to "conclusion"

Response: Change has been made accordingly.

Reviewer 4 Report

How much data was split into train, test, validation data??

Question is how that much data from 50 sites was collected as per your requirement? (availability and collection is the question)

Line 18-29: suggestion is to re-phrase this sentence “The negative environmental effects that come along with the usage of non-renewable energy, sky-high demand of electricity and the comparative ease in usage of the solar PV panels as a source of renewable energy contributes to the favouritism of solar PV panels”

Add reference to the following: Finally, collection and the processing of the meteorological data may be expensive and infeasible for a small-scale PV sites with low budget for operation.

Line 114-118: suggestion is to re-phrase this sentence “The flowchart of the overall steps performed in this study is shown in the Figure 1. In this section we describe the steps shown in the Figure 1. First we will discuss the first two steps: data collection and the data preprocessing. Then we will discuss the models that has been used in for the step of Figure 1. Finally, we will discuss the results obtained as part of the steps 4, 5 and 6 in Figure 1.

Line 281: Explain L2 in sentense “We train all of the used model with L2 loss while using ADAM”

Discussion and conclusion sections should be separated.

Author Response

I would like to thank the reviewer for spending their valuable time reviewing my manuscript. My responses to the comments are as follows:

Comment 1: How much data was split into train, test, validation data?

Response: The data was split into train, test, validation in the ratio: 8:1:1. This information has also been added to the manuscript.

Comment 2:  Question is how that much data from 50 sites was collected as per your requirement? (availability and collection is the question)

Response: There are a total of 24065 data points in total for each site collected over the period of one year. However, due to high irregularity in the later parts of the collected data only 11142 datapoints have been used. This information has now been added to the manuscript as well.

Comment 3: Line 18-29: suggestion is to re-phrase this sentence “The negative environmental effects that come along with the usage of non-renewable energy, sky-high demand of electricity and the comparative ease in usage of the solar PV panels as a source of renewable energy contributes to the favouritism of solar PV panels”

Response: The suggested change has been made.

Comment 4: Add reference to the following: Finally, collection and the processing of the meteorological data may be expensive and infeasible for a small-scale PV sites with low budget for operation.

Response: The suggested change has been made.

Comment 5: Line 114-118: suggestion is to re-phrase this sentence “The flowchart of the overall steps performed in this study is shown in the Figure 1. In this section we describe the steps shown in the Figure 1. First we will discuss the first two steps: data collection and the data preprocessing. Then we will discuss the models that has been used in for the step of Figure 1. Finally, we will discuss the results obtained as part of the steps 4, 5 and 6 in Figure 1.

Response: The suggested change has been made.

Comment 6: Line 281: Explain L2 in sentense “We train all of the used model with L2 loss while using ADAM”.

Response: The line has been further elaborated.

Comment 7: Discussion and conclusion sections should be separated.

Response: The change has been made as asked.

Round 2

Reviewer 1 Report

Thanks for your reply, and the paper has been improved a lot. However, there are a few concerns need to be discussed.

1) The data is split into train, validation and test set. How does the data is split with the 50 long horizon data (shown in Table 1). The former parts of all the sites as training set or some of the sites as training set? I don’t get.

2) The model sizes of RNN, GRU, LSTM and Transformer are different, so, will the performance be affected by the model sizes? For example, if the LSTM has the same model size with Transformer, will their performance be different?

3) In the multi-in multi-out setting (Figure 7), The FHL should be t, t+1, …, t+n.

Author Response

Reply to Reviewer 1

I would like to thank the reviewer for spending their valuable time reviewing my manuscript. My responses to the comments are as follows:

Comment 1: The data is split into train, validation and test set. How does the data is split with the 50 long horizon data (shown in Table 1). The former parts of all the sites as training set or some of the sites as training set? I don’t get.

Response: The Table 1 shows the initial raw data. The train, validation and test sets would consist of the data from same PV-sites in input and output, and they vary in terms of time stamps. Before the data is split into train, validation and test sets, however, the data needed to be pre-processed. The pre-processing was done differently for the 3 different setting (multi-in multi-out, multi-in uni-out and uni-in uni-out) and can be understood from Figure 7.

  • Multi-in multi-out setting: In each set of input and output, the input consists of data from N-sites of length equaling to ‘Lookback length’ and the output would consist of data from N-sites of length equaling to ‘Forecast Horizon Length.’
  • Multi-in uni-out setting: In each set of input and output data, the input consists of data from N-sites of length equaling to ‘Lookback Length’ and the output would consist of data from a single site (one of the sites used in the input as well) of length equaling to the ‘Forecast Horizon Length.’
  • Uni-in Uni-out setting: In each set of input and output data, the input would consist of data from a single site of length equaling to ‘Lookback length’ and the output would consist of data from same site of length equaling to the ‘Forecast horizon length.’

In each of the three settings, each set of input and output datapoints belong to continuous time stamp and the input and output time stamps do not overlap.

Comment 2: The model sizes of RNN, GRU, LSTM and Transformer are different, so, will the performance be affected by the model sizes? For example, if the LSTM has the same model size with Transformer, will their performance be different?

Response: The performance of each of the models would vary for different hyperparameters. So in our case, before performing the actual experiments, a set of initial experiments with different hyperparameters were performed to get reasonable results for each of the models. The best set of hyperparameters were then used for all of the experiments.

Comment 3: In the multi-in multi-out setting (Figure 7), The FHL should be t, t+1, …, t+n.Response: Explanation of the three settings have been added in section 2.3. Additionally, to avoid confusion, the settings have been renamed as i.e., multi-in multi-out setting, multi-in single-out setting and uni-in uni-out setting.

Response: The figure has been changed accordingly.

Reviewer 2 Report

Thank you very much for answering the questions, I have no further comments.

Author Response

I would like to thank the reviewer for spending their valuable time reviewing my manuscript. 

Reviewer 3 Report

The revised article is fine. 

Author Response

(The authors gave the same response as above.)
